# Implicit SVD for Graph Representation Learning

**Sami Abu-El-Haija** [*]
USC Information Sciences Institute
sami@haija.org

**Hesham Mostafa, Marcel Nassar**
Intel Labs
{hesham.mostafa,marcel.nassar}@intel.com

**Valentino Crespi, Greg Ver Steeg, Aram Galstyan**
USC Information Sciences Institute
{vcrespi,gregv,galstyan}@isi.edu

## Abstract

Recent improvements in the performance of state-of-the-art (SOTA) methods for Graph Representational Learning (GRL) have come at the cost of significant computational resource requirements for training, e.g., for calculating gradients via backprop over many data epochs. Meanwhile, Singular Value Decomposition (SVD) can find closed-form solutions to convex problems, using merely a handful of epochs. In this paper, we make GRL more computationally tractable for those with modest hardware. We design a framework that computes SVD of *implicitly* defined matrices, and apply this framework to several GRL tasks. For each task, we derive linear approximation of a SOTA model, where we design (expensive-to-store) matrix $\mathbf{M}$ and train the model, in closed-form, via SVD of $\mathbf{M}$, without calculating entries of $\mathbf{M}$. By converging to a unique point in one step, and without calculating gradients, our models show competitive empirical test performance over various graphs such as article citation and biological interaction networks. More importantly, SVD can initialize a deeper model, that is architected to be non-linear almost everywhere, though behaves linearly when its parameters reside on a hyperplane, onto which SVD initializes. The deeper model can then be fine-tuned within only a few epochs. Overall, our procedure trains hundreds of times faster than state-of-the-art methods, while competing on empirical test performance. We open-source our implementation at: https://github.com/samihaija/isvd

## 1   Introduction

Truncated Singular Value Decomposition (SVD) provides solutions to a variety of mathematical problems, including computing a matrix rank, its pseudo-inverse, or mapping its rows and columns onto the *orthonormal singular bases* for low-rank approximations. Machine Learning (ML) software frameworks (such as TensorFlow) offer efficient SVD implementations, as SVD can estimate solutions for a variaty of tasks, e.g., in *computer vision* [Turk and Pentland, 1991], *weather prediction* [Molteni et al., 1996], *recommendation* [Koren et al., 2009], *language* [Deerwester et al., 1990, Levy and Goldberg, 2014], and more-relevantly, *graph representation learning* (GRL) [Qiu et al., 2018].

SVD's benefits include training models, without calculating gradients, to arrive at globally-unique solutions, optimizing Frobenius-norm objectives (§2.3), without requiring hyperparameters for the learning process, such as the choice of the learning algorithm, step-size, regularization coefficient, etc. Typically, one *constructs* a **design matrix** $\mathbf{M}$, such that, its decomposition provides a solution to a task of interest. Unfortunately, existing popular ML frameworks [Abadi et al., 2016, Paszke et al., 2019] cannot calculate the SVD of an arbitrary linear matrix given its computation graph: they

---

[*]Part of this work was done during internship at Intel Labs.

35th Conference on Neural Information Processing Systems (NeurIPS 2021).

compute the matrix (entry-wise) then[2] its decomposition. This limits the scalability of these libraries in several cases of interest, such as in GRL, when explicit calculation of the matrix is prohibitive due to memory constraints. These limitations render SVD as impractical for achieving state-of-the-art (SOTA) for tasks at hand. This has been circumvented by Qiu et al. [2018] by sampling $\mathbf{M}$ entry-wise, but this produces sub-optimal estimation error and experimentally degrades the empirical test performance (§7: Experiments).

We design a software library that allows **symbolic** definition of $\mathbf{M}$, via composition of matrix operations, and we implement an SVD algorithm that can decompose $\mathbf{M}$ from said symbolic representation, without need to compute $\mathbf{M}$. This is valuable for many GRL tasks, where the design matrix $\mathbf{M}$ is too large, *e.g.*, quadratic in the input size. With our implementation, we show that SVD can perform learning, orders of magnitudes faster than current alternatives.

Currently, SOTA GRL models are generally graph neural networks trained to optimize cross-entropy objectives. Their inter-layer non-linearities place their (many) parameters onto a non-convex objective surface where convergence is rarely verified[3]. Nonetheless, these models can be *convexified* (§3) and trained via SVD, **if** we remove nonlinearities between layers **and** swap the cross-entropy objective with Frobenius norm minimization. Undoubtedly, such linearization incurs a drop of accuracy on empirical test performance. Nonetheless, we show that the (convexified) model's parameters learned by SVD can provide initialization to deeper (non-linear) models, which then can be fine-tuned on cross-entropy objectives. The non-linear models are endowed with our novel Split-ReLu layer, which has twice as many parameters as a ReLu fully-connected layer, and behaves as a linear layer when its parameters reside on some hyperplane (§5.2). Training on modest hardware (e.g., laptop) is sufficient for this learning pipeline (convexify $\rightarrow$ SVD $\rightarrow$ fine-tune) yet it trains much faster than current approaches, that are commonly trained on expensive hardware. We summarize our contributions as:

1. We open-source a flexible python software library that allows symbolic definition of matrices and computes their SVD without explicitly calculating them.

2. We linearize popular GRL models, and train them via SVD of design matrices.

3. We show that fine-tuning a few parameters on-top of the SVD initialization sets state-of-the-art on many GRL tasks while, overall, training orders-of-magnitudes faster.

## 2    Preliminaries & notation

We denote a graph with $n$ nodes and $m$ edges with an *adjacency matrix* $\mathbf{A} \in \mathbb{R}^{n \times n}$ and additionally, if nodes have ($d$-dimensional) features, with a *feature matrix* $\mathbf{X} \in \mathbb{R}^{n \times d}$. If nodes $i, j \in [n]$ are connected then $\mathbf{A}_{ij}$ is set to their edge weight and otherwise $\mathbf{A}_{ij} = 0$. Further, denote the (row-wise normalized) *transition matrix* as $\mathcal{T} = \mathbf{D}^{-1}\mathbf{A}$ and denote the symmetrically normalized adjacency with self-connections as $\widehat{\mathbf{A}} = (\mathbf{D} + \mathbf{I})^{-\frac{1}{2}}(\mathbf{A} + \mathbf{I})(\mathbf{D} + \mathbf{I})^{-\frac{1}{2}}$ where $\mathbf{I}$ is identity matrix.

We review model classes: (1) network embedding and (2) message passing that we define as follows. The first inputs a graph $(\mathbf{A}, \mathbf{X})$ and outputs *node embedding matrix* $\mathbf{Z} \in \mathbb{R}^{n \times z}$ with $z$-dimensions per node. $\mathbf{Z}$ is then used for an upstream task, *e.g.*, link prediction. The second class utilizes a function $\mathbf{H} : \mathbb{R}^{n \times n} \times \mathbb{R}^{n \times d} \rightarrow \mathbb{R}^{n \times z}$ where the function $\mathbf{H}(\mathbf{A}, \mathbf{X})$ is usually directly trained on the upstream task, *e.g.*, node classification. In general, the first class is transductive while the second is inductive.

### 2.1    Network embedding models based on DeepWalk & node2vec

The seminal work of DeepWalk [Perozzi et al., 2014] embeds nodes of a network using a two-step process: (i) simulate random walks on the graph – each walk generating a sequence of node IDs then (ii) pass the walks (node IDs) to a language word embedding algorithm, e.g. word2vec [Mikolov et al., 2013], as-if each walk is a sentence. This work was extended by node2vec [Grover and Leskovec, 2016] among others. It has been shown by Abu-El-Haija et al. [2018] that the learning outcome of

---

[2]TensorFlow can caluclate matrix-free SVD if one implements a `LinearOperator`, as such, our code could be re-implemented as a routine that can convert `TensorGraph` to `LinearOperator`.

[3]Practitioners rarely verify that $\nabla_\theta J = 0$, where $J$ is mean train objective and $\theta$ are model parameters.

the two-step process of DeepWalk is equivalent, in expectation, to optimizing a single objective[4]:

$$\min_{\mathbf{Z}=\{\mathbf{L},\mathbf{R}\}} \sum_{(i,j)\in[n]\times[n]} \left[ - \mathop{\mathbb{E}}_{q\sim Q} [\mathcal{T}^q] \circ \log \sigma(\mathbf{L}\mathbf{R}^\top) - \lambda(1 - \mathbf{A}) \circ \log(1 - \sigma(\mathbf{L}\mathbf{R}^\top)) \right]_{ij}, \quad (1)$$

where $\mathbf{L}, \mathbf{R} \in \mathbb{R}^{n \times \frac{z}{2}}$ are named by word2vec as the *input* and *output* embedding matrices, $\circ$ is Hadamard product, and the $\log(.)$ and the standard logistic $\sigma(.) = (1 + \exp(.))^{-1}$ are applied element-wise. The objective above is weighted cross-entropy where the (left) positive term weighs the dot-product $\mathbf{L}_i^\top \mathbf{R}_j$ by the (expected) number of random walks simulated from $i$ and passing through $j$, and the (right) negative term weighs non-edges $(1 - \mathbf{A})$ by scalar $\lambda \in \mathbb{R}_+$. The *context distribution $Q$* stems from step (ii) of the process. In particular, word2vec accepts hyperparameter *context window size $C$* for its stochasatic sampling: when it samples a *center token* (node ID), it then samples its *context tokens* that are up-to distance $c$ from the center. The integer $c$ is sampled from a coin flip uniform on the integers $[1, 2, \dots, C]$ – as detailed by Sec.3.1 of [Levy et al., 2015]. Therefore, $P_Q(q \mid C) \propto \frac{C-q+1}{C}$. Since $q$ has support on $[C]$, then $P_Q(q \mid C) = \left( \frac{2}{(C+1)C} \right) \frac{C-q+1}{C}$.

## 2.2 Message passing graph networks for (semi-)supervised node classification

We are also interested in a class of (message passing) graph network models taking the general form:

$$\text{for } l = 0, 1, \dots L: \quad \mathbf{H}^{(l+1)} = \sigma_l \left( g(\mathbf{A})\mathbf{H}^{(l)}\mathbf{W}^{(l)} \right); \quad \mathbf{H}^{(0)} = \mathbf{X}; \quad \mathbf{H} = \mathbf{H}^{(L)}; \quad (2)$$

where $L$ is the number of layers, $\mathbf{W}^{(l)}$'s are trainable parameters, $\sigma_l$'s denote element-wise activations (e.g. logistic or ReLu), and $g$ is some (possibly trainable) transformation of adjacency matrix. GCN [Kipf and Welling, 2017] set $g(\mathbf{A}) = \widehat{\mathbf{A}}$, GAT [Veličković et al., 2018] set $g(\mathbf{A}) = \mathbf{A} \circ$ MultiHeadedAttention and GIN [Xu et al., 2019] as $g(\mathbf{A}) = \mathbf{A} + (1 + \epsilon)\mathbf{I}$ with $\epsilon > 0$. For node classification, it is common to set $\sigma_L = \text{softmax}$ (applied row-wise), specify the size of $\mathbf{W}_L$ s.t. $\mathbf{H} \in \mathbb{R}^{n \times y}$ where $y$ is number of classes, and optimize cross-entropy objective:

$\min_{\{\mathbf{W}_j\}_{j=1}^L} [-\mathbf{Y} \circ \log \mathbf{H} - (1 - \mathbf{Y}) \circ \log(1 - \mathbf{H})]$, where $\mathbf{Y}$ is a binary matrix with one-hot rows indicating node labels. In semi-supervised settings where not all nodes are labeled, before measuring the objective, subset of rows can be kept in $\mathbf{Y}$ and $\mathbf{H}$ that correspond to labeled nodes.

## 2.3 Truncated Singular Value Decomposition (SVD)

SVD is an algorithm that approximates any matrix $\mathbf{M} \in \mathbb{R}^{r \times c}$ as a product of three matrices:

$$\text{SVD}_k(\mathbf{M}) \triangleq \arg\min_{\mathbf{U},\mathbf{S},\mathbf{V}} ||\mathbf{M} - \mathbf{U}\mathbf{S}\mathbf{V}^\top||_F \text{ subject to } \mathbf{U}^\top \mathbf{U} = \mathbf{V}^\top \mathbf{V} = \mathbf{I}_k; \ \mathbf{S} = \text{diag}(s_1, \dots, s_k).$$

The *orthonormal* matrices $\mathbf{U} \in \mathbb{R}^{r \times k}$ and $\mathbf{V} \in \mathbb{R}^{c \times k}$, respectively, are known as the left- and right-singular bases. The values along diagonal matrix $\mathbf{S} \in \mathbb{R}^{k \times k}$ are known as the *singular values*. Due to theorem of Eckart and Young [1936], SVD recovers the best rank-$k$ approximation of input $\mathbf{M}$, as measured by the Frobenius norm $||.||_F$. Further, if $k \geq \text{rank}(\mathbf{M}) \Rightarrow ||.||_F = 0$.

Popular SVD implementations follow Random Matrix Theory algorithm of Halko et al. [2009]. The prototype algorithm starts with a random matrix and repeatedly multiplies it by $\mathbf{M}$ and by $\mathbf{M}^\top$, interleaving these multiplications with orthonormalization. Our SVD implementation (in Appendix) also follows the prototype of [Halko et al., 2009], but with two modifications: (i) we replace the recommended orthonormalization step from QR decomposition to Cholesky decomposition, giving us significant computational speedups and (ii) our implementation accepts symbolic representation of $\mathbf{M}$ (§4), in lieu of its explicit value (constrast to TensorFlow and PyTorch, requiring explicit $\mathbf{M}$).

In §3, we derive linear first-order approximations of models reviewed in §2.1 & §2.2 and explain how SVD can train them. In §5, we show how they can be used as initializations of non-linear models.

---

[4]Derivation is in [Abu-El-Haija et al., 2018]. Unfortunately, matrix in Eq. 1 is dense with $\mathcal{O}(n^2)$ nonzeros.

# 3 Convex first-order approximations of GRL models

## 3.1 Convexification of Network Embedding Models

We can interpret objective 1 as self-supervised learning, since node labels are absent. Specifically, given a node $i \in [n]$, the task is to predict its neighborhood as weighted by the row vector $\mathbb{E}_q[\mathcal{T}^q]_i$, representing the subgraph[5] around $i$. Another interpretation is that Eq. 1 is a decomposition objective: multiplying the tall-and-thin matrices, as $\mathbf{LR}^\top \in \mathbb{R}^{n \times n}$, should give a larger value at $(\mathbf{LR}^\top)_{ij} = \mathbf{L}_j^\top \mathbf{R}_i$ when nodes $i$ and $j$ are well-connected but a lower value when $(i,j)$ is not an edge. We propose a matrix such that its decomposition can incorporate the above interpretations:

$$\widehat{\mathbf{M}}^{(\text{NE})} = \mathbb{E}_{q|C}[\mathcal{T}^q] - \lambda(1 - \mathbf{A}) = \left( \frac{2}{(C+1)C} \right) \sum_{q=1}^{C} \left( \frac{C - q + 1}{C} \right) \mathcal{T}^q - \lambda(1 - \mathbf{A}) \quad (3)$$

If nodes $i, j$ are nearby, share a lot of connections, and/or in the same community, then entry $\widehat{\mathbf{M}}^{(\text{NE})}_{ij}$ should be positive. If they are far apart, then $\widehat{\mathbf{M}}^{(\text{NE})}_{ij} = -\lambda$. To embed the nodes onto a low-rank space that approximates this information, one can decompose $\widehat{\mathbf{M}}^{(\text{NE})}$ into two thin matrices $(\mathbf{L}, \mathbf{R})$:

$$\mathbf{LR}^\top \approx \widehat{\mathbf{M}}^{(\text{NE})} \Longleftrightarrow (\mathbf{LR}^\top)_{i,j} = \langle \mathbf{L}_i, \mathbf{R}_j \rangle \approx \widehat{\mathbf{M}}^{(\text{NE})}_{ij} \quad \text{for all} \ \ i, j \in [n]. \quad (4)$$

SVD gives low-rank approximations that minimize the Frobenius norm of error (§2.3). The remaining challenge is computational burden: the right term $(1 - \mathbf{A})$, *a.k.a*, graph compliment, has $\approx n^2$ non-zero entries and the left term has non-zero at entry $(i, j)$ if nodes $i, j$ are within distance $C$ away, as $q$ has support on $[C]$ – for reference Facebook network has an average distance of 4 [Backstrom et al., 2012] i.e. yielding $\mathcal{T}^4$ with $\mathcal{O}(n^2)$ nonzero entries – Nonetheless, Section §4 presents a framework for decomposing $\widehat{\mathbf{M}}$ from its symbolic representation, without explicitly computing its entries. Before moving forward, we note that one can replace $\mathcal{T}$ in Eq. 3 by its symmetrically normalized counterpart $\widehat{\mathbf{A}}$, recovering a basis where $\mathbf{L} = \mathbf{R}$. This symmetric modeling might be emperically preferred for undirected graphs. **Learning** can be performed via SVD. Specifically, the node at the $i^{\text{th}}$ row and the node at the $j^{\text{th}}$th column will be embedded, respectively, in $\mathbf{L}_i$ and $\mathbf{R}_j$ computed as:

$$\mathbf{U}, \mathbf{S}, \mathbf{V} \leftarrow \text{SVD}_k(\widehat{\mathbf{M}}^{(\text{NE})}); \qquad \mathbf{L} \leftarrow \mathbf{US}^{\frac{1}{2}}; \qquad \mathbf{R} \leftarrow \mathbf{VS}^{\frac{1}{2}} \quad (5)$$

In this $k$-dim space of rows and columns, Euclidean measures are plausible: **Inference** of nodes' similarity at row $i$ and column $j$ can be modeled as $f(i, j) = \langle \mathbf{L}_i, \mathbf{R}_j \rangle = \mathbf{U}_i^\top \mathbf{S} \mathbf{V}_j \triangleq \langle \mathbf{U}_i, \mathbf{V}_j \rangle_{\mathbf{s}}$.

## 3.2 Convexification of message passing graph networks

Removing all $\sigma_l$'s from Eq. 2 and setting $g(\mathbf{A}) = \widehat{\mathbf{A}}$ gives outputs of layers 1, 2, and $L$, respectively,

$$\text{as:} \quad \widehat{\mathbf{A}}\mathbf{X}\mathbf{W}^{(1)} \quad , \quad \widehat{\mathbf{A}}^2\mathbf{X}\mathbf{W}^{(1)}\mathbf{W}^{(2)} \quad , \text{ and } \quad \widehat{\mathbf{A}}^L\mathbf{X}\mathbf{W}^{(1)}\mathbf{W}^{(2)}\ldots\mathbf{W}^{(L)}. \quad (6)$$

Without non-linearities, adjacent parameter matrices can be absorbed into one another. Further, the model output can concatenate all layers, like JKNets [Xu et al., 2018], giving final model output of:

$$\mathbf{H}^{(\text{NC})}_{\text{linearized}} = \begin{bmatrix} \mathbf{X} & \vdots & \widehat{\mathbf{A}}\mathbf{X} & \vdots & \widehat{\mathbf{A}}^2\mathbf{X} & \vdots & \ldots & \vdots & \widehat{\mathbf{A}}^L\mathbf{X} \end{bmatrix} \widehat{\mathbf{W}} \quad \triangleq \quad \widehat{\mathbf{M}}^{(\text{NC})}\widehat{\mathbf{W}}, \quad (7)$$

where the linearized model implicitly constructs design matrix $\widehat{\mathbf{M}}^{(\text{NC})} \in \mathbb{R}^{n \times F}$ and multiplies it with parameter $\widehat{\mathbf{W}} \in \mathbb{R}^{F \times y}$ – here, $F = d + dL$. Crafting design matrices is a creative process (§5.3). **Learning** can be performed by minimizing the Frobenius norm: $||\mathbf{H}^{(\text{NC})} - \mathbf{Y}||_{\text{F}} = ||\widehat{\mathbf{M}}^{(\text{NC})}\widehat{\mathbf{W}} - \mathbf{Y}||_{\text{F}}$. Moore-Penrose Inverse (a.k.a, the psuedoinverse) provides one such minimizer:

$$\widehat{\mathbf{W}}^* = \text{argmin}_{\widehat{\mathbf{W}}} \left|\left| \widehat{\mathbf{M}}\widehat{\mathbf{W}} - \mathbf{Y} \right|\right|_{\text{F}} = \widehat{\mathbf{M}}^\dagger \mathbf{Y} \approx \mathbf{VS}^+\mathbf{U}^\top\mathbf{Y}, \quad (8)$$

with $\mathbf{U}, \mathbf{S}, \mathbf{V} \leftarrow \text{SVD}_k(\widehat{\mathbf{M}})$. Notation $\mathbf{S}^+$ reciprocates non-zero entries of diagonal $\mathbf{S}$ [Golub and Loan, 1996]. Multiplications in the right-most term should, for efficiency, be executed right-to-left.

---

[5] $\mathbb{E}_q[\mathcal{T}^q]_i$ is a distribution on $[n]$: entry $j$ equals prob. of walk starting at $i$ ending at $j$ if walk length $\sim \mathcal{U}[C]$.

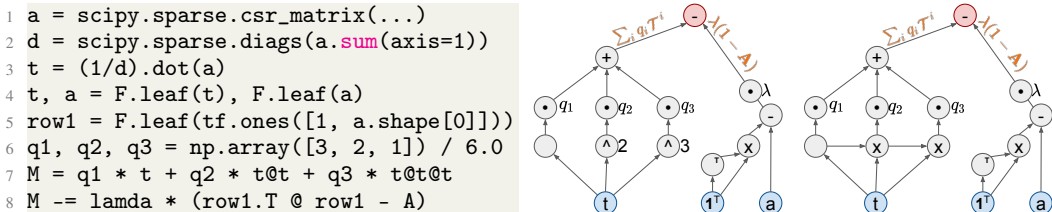

```
1  a = scipy.sparse.csr_matrix(...)
2  d = scipy.sparse.diags(a.sum(axis=1))
3  t = (1/d).dot(a)
4  t, a = F.leaf(t), F.leaf(a)
5  row1 = F.leaf(tf.ones([1, a.shape[0]]))
6  q1, q2, q3 = np.array([3, 2, 1]) / 6.0
7  M = q1 * t + q2 * t@t + q3 * t@t@t
8  M -= lamda * (row1.T @ row1 - A)
```

Figure 1: Symbolic Matrix Representation. **Left**: code using our framework to implicitly construct the design matrix $\mathbf{M} = \widehat{\mathbf{M}}^{(\mathrm{NE})}$ with our framework. **Center**: DAG corresponding to the code. **Right**: An equivalent automatically-optimized DAG (via lazy-cache, Fig. 2) requiring fewer floating point operations. The first 3 lines of code create explicit input matrices (that fit in memory): **a**djacency $\mathbf{A}$, diagonal **d**egree $\mathbf{D}$, and **t**ransition $\mathcal{T}$. Matrices are imported into our framework with `F.leaf` (depicted on computation DAGs in blue). Our classes overloads standard methods (`+`, `-`, `*`, `@`, `**`) to construct computation nodes (intermediate in grey). The output node (in red) needs not be exactly calculated yet can be efficiently multiplied by any matrix by recursive downward traversal.

The pseudoinverse $\widehat{\mathbf{M}}^{\dagger} \approx \mathbf{V}\mathbf{S}^{+}\mathbf{U}^{\top}$ recovers the $\widehat{\mathbf{W}}^{*}$ with least norm (§6, Theorem 1). The $\approx$ becomes $=$ when $k \geq \mathrm{rank}(\widehat{\mathbf{M}})$.

In semi-supervised settings, one can take rows subset of either (i) $\mathbf{Y}$ and $\mathbf{U}$, or of (ii) $\mathbf{Y}$ and $\mathbf{M}$, keeping only rows that correspond to labeled nodes. Option (i) is supported by existing frameworks (e.g., `tf.gather()`) and our symbolic framework (§4) supports (ii) by *implicit row (or column) gather* – i.e., calculating SVD of submatrix of $\mathbf{M}$ without explicitly computing $\mathbf{M}$ nor the submatrix. **Inference** over a (possibly new) graph $(\mathbf{A}, \mathbf{X})$ can be calculated by (i) (implicitly) creating the design matrix $\widehat{\mathbf{M}}$ corresponding to $(\mathbf{A}, \mathbf{X})$ then (ii) multiplying by the explicitly calculated $\widehat{\mathbf{W}}^{*}$. As explained in §4, $\widehat{\mathbf{M}}$ need not to be explicitly calculated for computing multiplications.

## 4 Symbolic matrix representation

To compute the SVD of any matrix $\mathbf{M}$ using algorithm prototypes presented by Halko et al. [2009], **it suffices to provide functions that can multiply arbitrary vectors** with $\mathbf{M}$ and $\mathbf{M}^{\top}$, and **explicit calculation** of $\mathbf{M}$ is **not required**. Our software framework can symbolically represent $\mathbf{M}$ as a directed acyclic graph (DAG) of computations. On this DAG, each node can be one of two kinds:

1. **Leaf node** (no incoming edges) that **explicitly** holds a matrix. Multiplications against leaf nodes are directly executed via an underlying math framework (we utilize TensorFlow).
2. **Symbolic node** that only **implicitly** represents a matrix as as a function of other DAG nodes. Multiplications are recursively computed, traversing incoming edges, until leaf nodes.

For instance, suppose leaf DAG nodes $\mathbf{M}_1$ and $\mathbf{M}_2$, respectively, explicitly contain row vector $\in \mathbb{R}^{1 \times n}$ and column vector $\in \mathbb{R}^{n \times 1}$. Then, their (symbolic) product DAG node $\mathbf{M} = \mathbf{M}_2 @ \mathbf{M}_1$ is $\in \mathbb{R}^{n \times n}$. Although storing $\mathbf{M}$ explicitly requires $\mathcal{O}(n^2)$ space, multiplications against $\mathbf{M}$ can remain within $\mathcal{O}(n)$ space if efficiently implemented as $\langle \mathbf{M}, . \rangle = \langle \mathbf{M}_2, \langle \mathbf{M}_1, . \rangle \rangle$. Figure 1 shows code snippet for composing DAG to represent symbolic node $\widehat{\mathbf{M}}^{(\mathrm{NE})}$ (Eq. 3), from leaf nodes initialized with in-memory matrices. Appendex lists symbolic nodes and their implementations.

## 5 SVD initialization for deeper models fine-tuned via cross-entropy

### 5.1 Edge function for network embedding as a (1-dimensional) Gaussian kernel

SVD provides decent solutions to link prediction tasks. Computing $\mathbf{U}, \mathbf{S}, \mathbf{V} \leftarrow \mathrm{SVD}(\mathbf{M}^{(\mathrm{NE})})$ is much faster than training SOTA models for link prediction, yet, simple edge-scoring function $f(i, j) = \langle \mathbf{U}_i, \mathbf{V}_j \rangle_{\mathbf{S}}$ yields competitive empirical (test) performance. We propose $f$ with $\theta = \{\mu, s\}$:

$$f_{\mu,s}(i,j) = \mathbb{E}_{x \sim \overline{\mathcal{N}}(\mu,s)} \langle \mathbf{U}_i, \mathbf{V}_j \rangle_{\mathbf{S}^x} = \mathbf{U}_i^{\top} \mathbb{E}_x[\mathbf{S}^x] \mathbf{V}_j = \mathbf{U}_i^{\top} \left( \int_{\Omega} \mathbf{S}^x \overline{\mathcal{N}}(x \mid \mu, s) \, dx \right) \mathbf{V}_j, \quad (9)$$

where $\overline{\mathcal{N}}$ is the truncated normal distribution (we truncate to $\Omega = [0.5, 2]$). The integral can be approximated by discretization and applying softmax (see §A.4). The parameters $\mu \in \mathbb{R}, \sigma \in \mathbb{R}_{>0}$ can be optimized on cross-entropy objective for link-prediction:

$$\min_{\mu,s} - \mathbb{E}_{(i,j) \in \mathbf{A}} \left[ \log \left( \sigma(f_{\mu,s}(i,j)) \right) \right] - k_{(n)} \mathbb{E}_{(i,j) \notin \mathbf{A}} \left[ \log \left( 1 - \sigma(f_{\mu,s}(i,j)) \right) \right], \qquad (10)$$

where the left- and right-terms, respectively, encourage $f$ to score high for edges, and the low for non-edges. $k_{(n)} \in \mathbb{N}_{>0}$ controls the ratio of negatives to positives per batch (we use $k_{(n)} = 10$). If the optimization sets $\mu = 1$ and $s \approx 0$, then $f$ reduces to no-op. In fact, we initialize it as such, and we observe that $f$ converges **within one epoch**, on graphs we experimented on. If it converges as $\mu < 1$, VS $\mu > 1$, respectively, then $f$ would effectively squash, VS enlarge, the spectral gap.

## 5.2 Split-ReLu (deep) graph network for node classification (NC)

$\widehat{\mathbf{W}}^*$ from SVD (Eq.8) can initialize an $L$-layer graph network with input: $\mathbf{H}^{(0)} = \mathbf{X}$, with:

$$\text{message passing (MP)} \qquad \mathbf{H}^{(l+1)} = \left[ \widehat{\mathbf{A}} \mathbf{H}^{(l)} \mathbf{W}^{(l)}_{(p)} \right]_+ - \left[ \widehat{\mathbf{A}} \mathbf{H}^{(l)} \mathbf{W}^{(l)}_{(n)} \right]_+, \qquad (11)$$

$$\text{output} \qquad \mathbf{H} = \sum_{l=0}^{l=L} \left[ \mathbf{H}^{(l)} \mathbf{W}^{(l)}_{(op)} \right]_+ - \left[ \mathbf{H}^{(l)} \mathbf{W}^{(l)}_{(on)} \right]_+ \qquad (12)$$

$$\text{initialize MP} \qquad \mathbf{W}^{(l)}_{(p)} \leftarrow \mathbf{I}; \mathbf{W}^{(l)}_{(n)} \leftarrow -\mathbf{I}; \qquad (13)$$

$$\text{initialize output} \qquad \mathbf{W}^{(l)}_{(op)} \leftarrow \widehat{\mathbf{W}}^*_{[dl \,:\, d(l+1)]}; \mathbf{W}^{(l)}_{(on)} \leftarrow -\widehat{\mathbf{W}}^*_{[dl \,:\, d(l+1)]}; \qquad (14)$$

Element-wise $[.]_+ = \max(0, .)$. Further, $\mathbf{W}_{[i \,:\, j]}$ denotes rows from $(i)^{\text{th}}$ until $(j-1)^{\text{th}}$ of $\mathbf{W}$.

The deep network layers (Eq. 11&12) use our Split-ReLu layer which we formalize as:

$$\text{SplitReLu}(\mathbf{X}; \mathbf{W}_{(p)}, \mathbf{W}_{(n)}) = \left[ \mathbf{X} \mathbf{W}_{(p)} \right]_+ - \left[ \mathbf{X} \mathbf{W}_{(n)} \right]_+, \qquad (15)$$

where the subtraction is calculated entry-wise. The layer has twice as many parameters as standard fully-connected (FC) ReLu layer. In fact, learning algorithms can recover FC ReLu from SplitReLu by assigning $\mathbf{W}_{(n)} = 0$. More importantly, the layer behaves as linear in $\mathbf{X}$ when $\mathbf{W}_{(p)} = -\mathbf{W}_{(n)}$. On this hyperplane, this linear behavior allows us to establish the equivalency: the (non-linear) model $\mathbf{H}$ is equivalent to the linear $\mathbf{H}_{\text{linearized}}$ at initialization (Eq. 13&14) due to Theorem 2. Following the initialization, model can be fine-tuned on cross-entropy objective as in §2.2.

## 5.3 Creative Add-ons for node classification (NC) models

**Label re-use** (LR): Let $\widehat{\mathbf{M}}^{(NC)}_{LR} \triangleq \left[ \widehat{\mathbf{M}}^{(NC)} \;\vdots\; (\widehat{\mathbf{A}} - (\mathbf{D} + \mathbf{I})^{-1}) \mathbf{Y}_{[\text{train}]} \;\vdots\; (\widehat{\mathbf{A}} - (\mathbf{D} + \mathbf{I})^{-1})^2 \mathbf{Y}_{[\text{train}]} \right]$.
This follows the motivation of Wang and Leskovec [2020], Huang et al. [2021], Wang [2021] and their empirical results on ogbn-arxiv dataset, where $\mathbf{Y}_{[\text{train}]} \in \mathbb{R}^{n \times y}$ contains one-hot vectors at rows corresponding to labeled nodes but contain zero vectors for unlabeled (test) nodes. Our scheme is similar to concatenating $\mathbf{Y}_{[\text{train}]}$ into $\mathbf{X}$, but with care to prevent label leakage from row $i$ of $\mathbf{Y}$ to row $i$ of $\widehat{\mathbf{M}}$, as we zero-out the diagonal of the adjacency multiplied by $\mathbf{Y}_{[\text{train}]}$.

**Pseudo-Dropout** (PD): Dropout [Srivastava et al., 2014] reduces overfitting of models. It can be related to *data augmentation*, as each example is presented multiple times. At each time, it appears with a different set of *dropped-out features* – input or latent feature values, chosen at random, get replaced with zeros. As such, we can replicate the design matrix as: $\widehat{\mathbf{M}}^\top \leftarrow \left[ \widehat{\mathbf{M}}^\top \;\vdots\; \text{PD}(\widehat{\mathbf{M}})^\top \right]$. This row-wise concatenation maintains the width of $\widehat{\mathbf{M}}$ and therefore the number of model parameters.

In the above add-ons, concatenations, as well as PD, can be implicit or explicit (see §A.3).

Table 1: Dataset Statistics

| Dataset | Nodes | Edges | Source | Task | X |
|---|---|---|---|---|---|
| PPI | 3,852 proteins | 20,881 chem. interactions | node2vec | LP | ✗ |
| FB | 4,039 users | 88,234 friendships | SNAP | LP | ✗ |
| AstroPh | 17,903 researchers | 197,031 co-authorships | SNAP | LP | ✗ |
| HepTh | 8,638 researchers | 24,827 co-authorships | SNAP | LP | ✗ |
| Cora | 2,708 articles | 5,429 citations | Planetoid | SSC | ✓ |
| Citeseer | 3,327 articles | 4,732 citations | Planetoid | SSC | ✓ |
| Pubmed | 19,717 articles | 44,338 citations | Planetoid | SSC | ✓ |
| ogbn-ArXiv | 169,343 papers | 1,166,243 citations | OGB | SSC | ✓ |
| ogbl-DDI | 4,267 drugs | 1,334,889 interactions | OGB | LP | ✓ |

## 6  Analysis & Discussion

**Theorem 1.** (Min. Norm) *If system $\widehat{\mathbf{M}}\widehat{\mathbf{W}} = \mathbf{Y}$ is underdetermined[6] with rows of $\widehat{\mathbf{M}}$ being linearly independent, then solution space $\widehat{\mathcal{W}}^* = \left\{ \widehat{\mathbf{W}} \,\middle|\, \widehat{\mathbf{M}}\widehat{\mathbf{W}} = \mathbf{Y} \right\}$ has infinitely many solutions. Then, for $k \geq rank(\widehat{\mathbf{M}})$, matrix $\widehat{\mathbf{W}}^*$, recovered by Eq.8 satisfies: $\widehat{\mathbf{W}}^* = \arg\min_{\widehat{\mathbf{W}} \in \widehat{\mathcal{W}}^*} ||\widehat{\mathbf{W}}||_F^2$.*

Theorem 1 implies that, even though one can design a wide $\widehat{\mathbf{M}}^{(\text{NC})}$ (Eq.7), *i.e.*, with many layers, the recovered parameters with least norm should be less prone to overfitting. Recall that this is the goal of L2 regularization. Analysis and proofs are in the Appendix.

**Theorem 2.** (Non-linear init) *The initialization Eq. 13&14 yields* $\mathbf{H}^{(\text{NC})}_{\text{linearized}} = \mathbf{H}\big|_{\theta \leftarrow \text{via Eq. 13\&14}}$.

Theorem 2 implies that the deep (nonlinear) model is the same as the linear model, at the initialization of $\theta$ (per Eq. 13&14, using $\widehat{\mathbf{W}}^*$ as Eq. 8). Cross-entropy objective can then fine-tune $\theta$.

This end-to-end process, of (i) computing SVD bases and (ii) training the network $f_\theta$ on singular values, *advances* SOTA on competitve benchmarks, with (i) converging (quickly) to a unique solution and (ii) containing merely a few parameters $\theta$ – see §7.

## 7  Applications & Experiments

We download and experiment on 9 datasets summarized in Table 1.

We attempt link prediction (LP) tasks on smaller graph datasets (< 1 million edges) of: Protein-Protein Interactions (PPI) graph from Grover and Leskovec [2016]; as well as ego-Facebook (FB), AstroPh, HepTh from Stanford SNAP [Leskovec and Krevl, 2014]. For these datasets, we use the train-test splits of Abu-El-Haija et al. [2018]. We also attempt semi-supervised node classification (SSC) tasks on smaller graphs of Cora, Citeseer, Pubmed, all obtained from Planetoid [Yang et al., 2016]. For these smaller datasets, we only train and test using the SVD basis (without finetuning)

Further, we attempt on slightly-larger datasets (> 1 million edges) from Stanford's Open Graph Benchmark [OGB, Hu et al., 2020]. We use the official train-test-validation splits and evaluator of OGB. We attempt LP and SSC, respectively, on Drug Drug Interactions (ogbl-DDI) and ArXiv citation network (ogbn-ArXiv). For these larger datasets, we use the SVD basis as an initialization that we finetune, as described in §5. For time comparisons, we train all models on Tesla K80.

### 7.1  Test performance & runtime on smaller datasets from: Planetoid & Stanford SNAP

For SSC over Planetoid's datasets, both $\mathbf{A}$ and $\mathbf{X}$ are given. Additionally, only a handful of nodes are labeled. The goal is to classify the unlabeled test nodes. Table 2 summarizes the results. For **baselines**, we download code of GAT [Veličković et al., 2018], MixHop [Abu-El-Haija et al., 2019], GCNII [Chen et al., 2020] and re-ran them with instrumentation to record training time. However, for baselines Planetoid [Yang et al., 2016] and GCN [Kipf and Welling, 2017], we copied numbers

---

[6]E.g., if the number of labeled examples i.e. height of $\mathbf{M}$ and $\mathbf{Y}$ is smaller than the width of $\mathbf{M}$.

Table 2: Test accuracy (& train time) on citation graphs for task: *semi-supervised node classification*.

| Graph dataset: | Cora | | Citeseer | | Pubmed | |
|---|---|---|---|---|---|---|
| **Baselines:** | acc | tr.time | acc | tr.time | acc | tr.time |
| Planetoid | 75.7 | (13s) | 64.7 | (26s) | 77.2 | (25s) |
| GCN | 81.5 | (4s) | 70.3 | (7s) | 79.0 | (83s) |
| GAT | 83.2 | (1m) | 72.4 | (3m) | 77.7 | (6m) |
| MixHop | 81.9 | (26s) | 71.4 | (31s) | 80.8 | (1m) |
| GCNII | 85.5 | (2m) | 73.4 | (3m) | 80.3 | (2m) |
| **Our models:** | acc | tr.time | acc | tr.time | acc | tr.time |
| $\text{iSVD}_{100}(\widehat{\mathbf{M}}^{(\text{NC})})$ | 82.0 ±0.13 | (0.1s) | 71.4 ±0.22 | (0.1s) | 78.9 ±0.31 | (0.3s) |
| + dropout (§5.3) | 82.5 ±0.46 | (0.1s) | 71.5 ±0.53 | (0.1s) | 78.9 ±0.59 | (0.2s) |

Table 3: Test ROC-AUC (& train time) on Stanford SNAP graphs for task: *link prediction*.

| Graph dataset: | FB | | AstroPh | | HepTh | | PPI | |
|---|---|---|---|---|---|---|---|---|
| **Baselines:** | AUC | tr.time | AUC | tr.time | AUC | tr.time | AUC | tr.time |
| WYS | 99.4 | (54s) | 97.9 | (32m) | 93.6 | (4m) | 89.8 | (46s) |
| n2v | 99.0 | (30s) | 97.8 | (2m) | 92.3 | (55s) | 83.1 | (27s) |
| NetMF | 97.6 | (5s) | 96.8 | (9m) | 90.5 | (72s) | 73.6 | (7s) |
| $\widetilde{\text{NetMF}}$ | 97.0 | (4s) | 81.9 | (4m) | 85.0 | (48s) | 63.6 | (10s) |
| **Our models:** | AUC | tr.time | AUC | tr.time | AUC | tr.time | AUC | tr.time |
| $\text{iSVD}_{32}(\widehat{\mathbf{M}}^{(\text{NE})})$ | 99.1 ±1e-6 | (0.2s) | 94.4 ±4e-4 | (0.5s) | 90.5 ±0.1 | (0.1s) | 89.3 ±0.01 | (0.1s) |
| $\text{iSVD}_{256}(\widehat{\mathbf{M}}^{(\text{NE})})$ | 99.3 ±9e-6 | (2s) | 98.0 ±0.01 | (7s) | 90.1 ±0.54 | (2s) | 89.3 ±0.48 | (1s) |

from [Kipf and Welling, 2017]. For **our models**, the row labeled $\text{iSVD}_{100}(\widehat{\mathbf{M}}^{(\text{NC})})$, we run our implicit SVD twice per graph. The first run incorporates structural information: we (implicitly) construct $\widehat{\mathbf{M}}^{(\text{NE})}$ with $\lambda = 0.05$ and $C = 3$, then obtain $\mathbf{L}, \mathbf{R} \leftarrow \text{SVD}_{64}(\widehat{\mathbf{M}}^{(\text{NE})})$, per Eq. 5. Then, we concatenate $\mathbf{L}$ and $\mathbf{R}$ into $\mathbf{X}$. Then, we PCA the resulting matrix to 1000 dimensions, which forms our new $\mathbf{X}$. The second SVD run is to train the classification model parameters $\widehat{\mathbf{W}}^*$. From the PCA-ed $\mathbf{X}$, we construct the implicit matrix $\widehat{\mathbf{M}}^{(\text{NC})}$ with $L = 15$ layers and obtain $\widehat{\mathbf{W}}^* = \mathbf{V}\mathbf{S}^+\mathbf{U}^\top\mathbf{Y}_{[\text{train}]}$ with $\mathbf{U}, \mathbf{S}, \mathbf{V} \leftarrow \text{SVD}_{100}(\widehat{\mathbf{M}}^{(\text{NC})}_{[\text{train}]})$, per in RHS of Eq. 8. For our second "+ dropout" model variant, we (implicit) augment the data by $\widehat{\mathbf{M}}^{(\text{NC})\top} \leftarrow \left[\widehat{\mathbf{M}}^{(\text{NC})\top} \mid \text{PD}(\widehat{\mathbf{M}}^{(\text{NC})})^\top\right]$, update indices $[\text{train}] \leftarrow \left[\text{train} \mid \text{train}\right]^\top$ then similarly learn as: $\text{SVD}_{100}(\widehat{\mathbf{M}}^{(\text{NC})}_{[\text{train}]}) \rightarrow \widehat{\mathbf{W}}^* = \mathbf{V}\mathbf{S}^+\mathbf{U}^\top\mathbf{Y}_{[\text{train}]}$
**Discussion:** Our method is competitive yet trains faster than SOTA. In fact, the only method that reliably beats ours on all dataset is GCNII, but its training time is about one-thousand-times longer.

For LP over SNAP and node2vec datasets, the training adjacency $\mathbf{A}$ is given but not $\mathbf{X}$. The split includes test positive and negative edges, which are used to measure a ranking metric: ROC-AUC, where the metric increases when test positive edges are ranked above than negatives. Table 3 summarizes the results. For **baselines**, we download code of WYS [Abu-El-Haija et al., 2018]; we use the efficient implementation of PyTorch-Geometric [Fey and Lenssen, 2019] for node2vec (n2v) and we download code of Qiu et al. [2018] and run it with their two variants, denoting their first variant as NetMF (for *exact*, explicitly computing the design matrix) as and their second variant as $\widetilde{\text{NetMF}}$ (for *approximate*, sampling matrix entry-wise) – their code runs SVD after computing either variant. For the first of **our models**, we compute $\mathbf{U}, \mathbf{S}, \mathbf{V} \leftarrow \text{SVD}_{32}(\widehat{\mathbf{M}}^{(\text{NE})})$ and score every test edge as $\mathbf{U}_i^\top \mathbf{S}\mathbf{V}_j$. For the second, we first run $\text{SVD}_{256}$ on **half** of the training edges, determine the "*best*" rank $\in \{8, 16, 32, 128, 256\}$ by measuring the AUC on the remaining half of training edges, then using this best rank, recompute the SVD on the entire training set, then finally score the test edges. **Discussion**: Our method is competitive on SOTA while training much faster. Both NetMF and WYS explicitly calculate a dense matrix before factorization. On the other hand, $\widetilde{\text{NetMF}}$ approximates the matrix entry-wise, trading accuracy for training time. In our case, we have the best

Table 4: Test Hits@20 for link prediction over Drug-Drug Interactions Network (ogbl-ddi).

| Graph dataset: | ogbl-DDI | | |
|---|---|---|---|
| **Baselines:** | | HITS@20 | tr.time |
| DEA+JKNet [Yang et al., 2021] | | 76.72 ±2.65 | (60m) |
| LRGA+n2v [Hsu and Chen, 2021] | | 73.85 ±8.71 | (41m) |
| MAD [Luo et al., 2021] | | 67.81 ±2.94 | (2.6h) |
| LRGA+GCN [Puny et al., 2020] | | 62.30 ±9.12 | (10m) |
| GCN+JKNet [Xu et al., 2018] | | 60.56 ±8.69 | (21m) |
| **Our models:** | | HITS@20 | tr.time |
| (a) iSVD$_{100}(\widehat{\mathbf{M}}^{(NE)})$ | (§3.1, Eq. 5) | 67.86 ±0.09 | (6s) |
| (b) + finetune $f_{\mu,s}(\mathbf{S})$ | (§5.1, Eq. 9 & 10; sets $\mu = 1.15$) | 79.09 ±0.18 | (24s) |
| (c) + update $\mathbf{U,S,V}$ | (on validation, keeps $f_{\mu,s}$ fixed) | 84.09 ±0.03 | (30s) |

of both worlds: using our symbolic representation and SVD implementation, we can decompose the design matrix while only implicitly representing it, as good as if we had explicitly calculated it.

## 7.2 Experiments on Stanford's OGB datasets

We summarize experiments ogbl-DDI and ogbn-ArXiv, respectively, in Tables 4 and 5. For **baselines**, we copy numbers from the public leaderboard, where the competition is fierce. We then follow links on the leaderboard to download author's code, that we re-run, to measure the training time. **For our models on ogbl-DDI,** we (a) first calculate $\mathbf{U}, \mathbf{S}, \mathbf{V} \leftarrow \text{SVD}_{100}(\widehat{\mathbf{M}}^{(NE)})$ built only from training edges and score test edge $(i, j)$ using $\mathbf{U}_i^\top \mathbf{S} \mathbf{V}_j$. Then, we (b) then finetune $f_{\mu,s}$ (per §5.1, Eq. 9 & 10)) for **only a single epoch** and score using $f_{\mu,s}(i, j)$. Then, we (c) update the SVD basis to include edges from the validation partition and also score using $f_{\mu,s}(i, j)$. We report the results for the three steps. For the last step, the rules of OGB allows using the validation set for training, but only after the hyperparameters are finalized. The SVD has no hyperparameters (except the rank, which was already determined by the first step). More importantly, this simulates a realistic situation: it is cheaper to obtain SVD (of an implicit matrix) than back-propagate through a model. For a time-evolving graph, one could run the SVD more often than doing gradient-descent epochs on a model. For **our models on ogbn-ArXiv**, we (a) compute $\mathbf{U}, \mathbf{S}, \mathbf{V} \leftarrow \text{SVD}_{250}(\widehat{\mathbf{M}}_{LR}^{(NC)})$ where the implicit matrix is defined in §5.3. We (b) repeat this process where we replicate $\widehat{\mathbf{M}}_{LR}^{(NC)}$ once: in the second replica, we replace the $\mathbf{Y}_{[train]}$ matrix with zeros (as-if, we drop-out the label with 50% probability). We (c) repeat the process where we concatenate two replicas of $\widehat{\mathbf{M}}_{LR}^{(NC)}$ into the design matrix, each with different dropout seed. We (d) fine-tune the last model over 15 epochs using stochastic GTTF [Markowitz et al., 2021]. **Discussion**: Our method competes or sets SOTA, while training much faster.
**Time improvements**: We replaced the recommended orthonormalization of [Halko et al., 2009] from QR decomposition, to Cholesky decomposition (§A.2). Further, we implemented *caching* to avoid computing sub-expressions if already calculated (§A.3.4). Speed-ups are shown in Fig. 2.

Table 5: Test classification accuracy over ogbn-arxiv.

| Graph dataset: | ogbn-ArXiv | |
|---|---|---|
| **Baselines:** | accuracy | tr.time |
| GAT+LR+KD [Ren, 2020] | 74.16 ±0.08 | (6h) |
| GAT+LR [Niu, 2020] | 73.99 ±0.12 | (3h) |
| AGDN [Sun and Wu, 2020] | 73.98 ±0.09 | (50m) |
| GAT+C&S [Huang et al., 2021] | 73.86 ±0.14 | (2h) |
| GCNII [Chen et al., 2020] | 72.74 ±0.16 | (3h) |
| **Our models:** | accuracy | tr.time |
| (a) iSVD$_{250}(\widehat{\mathbf{M}}_{LR}^{(NC)})$ (§3.2, Eq. 8) | 68.90 ±0.02 | (1s) |
| (b) + dropout(LR) (§5.3) | 69.34 ±0.02 | (3s) |
| (c) + dropout($\widehat{\mathbf{M}}_{LR}^{(NC)}$) (§5.3) | 71.95 ±0.03 | (6s) |
| (d) + finetune $\mathbf{H}$ (§5.2, Eq. 11-14) | 74.14 ±0.05 | (2m) |

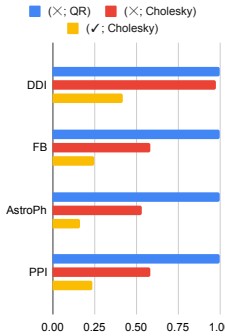

Figure 2: SVD runtime configs of (lazy caching; orthonormalization) as a ratio of SVD's common default (QR decomposition)

# 8 Related work

**Applications:** SVD was used to project rows & columns of matrix $\mathbf{M}$ onto an embedding space. $\mathbf{M}$ can be the Laplacian of a homogenous graph [Belkin and Niyogi, 2003], Adjacency of user-item bipartite graph [Koren et al., 2009], or stats [Deerwester et al., 1990, Levy and Goldberg, 2014] for word-to-document. We differ: our $\mathbf{M}$ is a *function* of (leaf) matrices – useful when $\mathbf{M}$ is expensive to store (e.g., quadratic). While Qiu et al. [2018] circumvents this by entry-wise sampling the (otherwise $n^2$-dense) $\mathbf{M}$, our SVD implementation could decompose exactly $\mathbf{M}$ without calculating it.

**Symbolic Software Frameworks:** including Theano [Al-Rfou et al., 2016], TensorFlow [Abadi et al., 2016] and PyTorch [Paszke et al., 2019], allow chaining operations to compose a computation (directed acyclic) graph (DAG). They can efficiently **run the DAG upwards** by evaluating (all entries of) matrix $\mathbf{M} \in \mathbb{R}^{r \times c}$ at any DAG node. Our DAG differs: instead of calculating $\mathbf{M}$, it provides product function $u_{\mathbf{M}}(.) = \mathbf{M} \times .$ – The **graph is run downwards** (reverse direction of edges).

**Matrix-free SVD:** For many matrices of interest, multiplying against $\mathbf{M}$ is computationally cheaper than explicitly storing $\mathbf{M}$ entry-wise. As such, many researchers implement $\text{SVD}(u_{\mathbf{M}})$, e.g. Calvetti et al. [1994], Wu and Simon [2000], Bose et al. [2019]. We differ in the programming flexibility: the earlier methods expect the practitioner to directly implement $u_{\mathbf{M}}$. On the other hand, our framework allows composition of $\mathbf{M}$ via operations native to the practitioner (e.g., @, +, concat), and $u_{\mathbf{M}}$ is automatically defined.

**Fast Graph Learning:** We have applied our framework for fast graph learning, but so as *sampling-based approaches* including [Chen et al., 2018, Chiang et al., 2019, Zeng et al., 2020, Markowitz et al., 2021]. We differ in that ours can be used to obtain an initial closed-form solution (very quickly) and can be fine-tuned afterwards using any of the aforementioned approaches. Additionally, Graphs shows one general application. Our framework might be useful in other areas utilizing SVD.

# 9 Conclusion, our limitations & possible negative societal impact

We develop a software framework for symbolically representing matrices and compute their SVD. Without computing gradients, this trains convexified models over GRL tasks, showing empirical metrics competitive with SOTA while training significantly faster. Further, convexified model parameters can initialize (deeper) neural networks that can be fine-tuned with cross entropy. Practitioners adopting our framework would now spend more effort in crafting design matrices instead of running experiments or tuning hyperparameters of the learning algorithm. We hope our framework makes high performance graph modeling more accessible by reducing reliance on energy-intensive computation.

**Limitations of our work**: From a representational prospective, **our space of functions is smaller** than TensorFlow's, as our DAG must be a linear transformation of its input matrices, *e.g.*, unable to encode element-wise transformations and hence demanding first-order linear approximation of models. As a result, our **gradient-free learning** can be performed using SVD. Further, our framework only works when the leaf nodes (*e.g.*, sparse adjacency matrix) fit in memory of one machine.

**Possible societal impacts**: First, our convexified models directly learn parameters in the feature space i.e. they are more explainable than deeper counterparts. Explainability is a double-edged sword: it gives better ability to interpret the model's behavior, but also allows for malicious users, e.g., to craft attacks on the system (if they can replicate the model parameters). Further, we apply our method on graphs. It is possible to train our models to detect sensitive attributes of social networks (e.g., ethnicity). However, such ethical concerns exists with any modeling technique over graphs.

# Acknowledgments and Disclosure of Funding

This material is based upon work supported by the Defense Advanced Research Projects Agency (DARPA) and the Army Contracting Command- Aberdeen Proving Grounds (ACC-APG) under Contract Number W911NF-18-C-0020.

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
