| $\mathrm{iSVD}_{32}(\widehat{\mathbf{M}}^{(\mathrm{NE})})$ | 99.1 ±1e-6 | (0.2s) | 94.4 ±4e-4 | (0.5s) | 90.5 ±0.1 | (0.1s) | 89.3 ±0.01 | (0.1s) |
| $\mathrm{iSVD}_{256}(\widehat{\mathbf{M}}^{(\mathrm{NE})})$ | 99.3 ±9e-6 | (2s) | 98.0 ±0.01 | (7s) | 90.1 ±0.54 | (2s) | 89.3 ±0.48 | (1s) |

from [Kipf and Welling, 2017]. For **our models**, the row labeled $\mathrm{iSVD}_{100}(\widehat{\mathbf{M}}^{(\mathrm{NC})})$, we run our implicit SVD twice per graph. The first run incorporates structural information: we (implicitly) construct $\widehat{\mathbf{M}}^{(\mathrm{NE})}$ with $\lambda = 0.05$ and $C = 3$, then obtain $\mathbf{L}, \mathbf{R} \leftarrow \mathrm{SVD}_{64}(\widehat{\mathbf{M}}^{(\mathrm{NE})})$, per Eq. 5. Then, we concatenate $\mathbf{L}$ and $\mathbf{R}$ into $\mathbf{X}$. Then, we PCA the resulting matrix to 1000 dimensions, which forms our new $\mathbf{X}$. The second SVD run is to train the classification model parameters $\widehat{\mathbf{W}}^*$. From the PCA-ed $\mathbf{X}$, we construct the implicit matrix $\widehat{\mathbf{M}}^{(\mathrm{NC})}$ with $L = 15$ layers and obtain $\widehat{\mathbf{W}}^* = \mathbf{V}\mathbf{S}^+\mathbf{U}^\top\mathbf{Y}_{[\mathrm{train}]}$ with $\mathbf{U}, \mathbf{S}, \mathbf{V} \leftarrow \mathrm{SVD}_{100}(\widehat{\mathbf{M}}^{(\mathrm{NC})}_{[\mathrm{train}]})$, per in RHS of Eq. 8. For our second "+ dropout" model variant, we (implicit) augment the data by $\widehat{\mathbf{M}}^{(\mathrm{NC})\top} \leftarrow [\widehat{\mathbf{M}}^{(\mathrm{NC})\top} \mid \mathrm{PD}(\widehat{\mathbf{M}}^{(\mathrm{NC})})^\top]$, update indices $[\mathrm{train}] \leftarrow [\mathrm{train} \mid \mathrm{train}]^\top$ then similarly learn as: $\mathrm{SVD}_{100}(\widehat{\mathbf{M}}^{(\mathrm{NC})}_{[\mathrm{

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

# A Appendix

## A.1 Hyperparameters

We list our hyperparameters so that others can replicate our results. Nonetheless, our code has a `README.md` file with instructions on how to run our code per dataset.

### A.1.1 Implicit matrix hyperparameters

**Planetoid datasets** [Yang et al., 2016] (Cora, CiteSeer, PubMed): we (implicitly) represent $\widehat{\mathbf{M}}^{(\text{NE})}$ with negative coefficient $\lambda = 0.05$ and context window $C = 3$; we construct $\widehat{\mathbf{M}}^{(\text{NC})}$ with number of layers $L = 15$. In one option, we use no dropout (first line of "our models" in Table 2) and in another (second line), we use dropout as described in §5.3.

**SNAP and node2vec datasets** (PPI, FB, AstroPh, HepTh): we (implicitly) represent $\widehat{\mathbf{M}}^{(\text{NE})}$ with negative coefficient $\lambda = 0.02$ and context window $C = 10$.

**Stanford OGB Drug-Drug Interactions** (ogbl-DDI): we (implicitly) represent $\widehat{\mathbf{M}}^{(\text{NE})}$ with negative coefficient $\lambda = 1$ and context window $C = 5$.

**Stanford OGB ArXiv** (ogbn-ArXiv): Unlike Planetoid datasets, we see in arXiv that appending the decomposition of $\widehat{\mathbf{M}}^{(\text{NE})}$ onto $\mathbf{X}$ does not improve the validation performance. Nonetheless, we construct $\widehat{\mathbf{M}}^{(\text{NC})}$ with $L = 2$: we apply two forms of dropout, as explained in §7.2.

**Justifications**: For Stanford OGB datasets, we make the selection based on the validation set. For the other (smaller) datasets, we choose them based on one datasets in each scenario (Cora for node classification and PPI for link prediction) and keep them fixed, for all datasets in the scenario – hyperparameter tuning (on validation) per dataset may increase test performance, however, our goal is to make a general framework that helps across many GRL tasks, more-so than getting bold on specific tasks.

### A.1.2 Finetuning hyperparameters

For Stanford OGB datasets (since competition is challenging), we use SVD to initialize a network that is finetuned on cross-entropy objectives. For ogbl-DDI, we finetune model in §5.1 using Adam optimizer with 1,000 positives and 10,000 negatives per batch, over one epoch, with learning rate of $1e^{-2}$. For ogbn-ArXiv, we finetune model in §5.2 using Adam optimizer using GTTF [Markowitz et al., 2021] with `fanouts=[4, 4]` (i.e., for every labeled node in batch, GTTF samples 4 neighbors, and 4 of their neighbors). With batch size 500, we train with learning rate $= 1e^{-3}$ for 7 epochs and with learning rate $= 1e^{-4}$ for 8 epochs (*i.e.*, 15 epochs total).

## A.2 SVD Implementation

The prototype algorithm for SVD described by Halko et al. [2009] suggests using QR-decomposition to for orthonormalizing[7] but we rather use Cholesky decomposition which is faster to calculate (our framework is written on top of TensorFlow 2.0 [Abadi et al., 2016]). Fig. 2 in the main paper shows the time comparison.

## A.3 Symbolic representation

Each node is a python class instance, that inherits base-class `SymbolicPF` (where PF stands for *product function*. Calculating SVD of (implicit) matrix $\mathbf{M} \in \mathbb{R}^{r \times c}$ is achievable, without explicitly knowing entries of $\widehat{\mathbf{M}}$, if we are able to multiply $\widehat{\mathbf{M}}$ by arbitrary $\mathbf{G}$.

1. `shape()` property must return a tuple of the shape of the underlying implicit matrix e.g. $= (r, c)$.

2. `dot(G)` must right-multiply an arbitrary explicit matrix $\mathbf{G} \in \mathbb{R}^{c \times \cdot}$ with (implicit) $\widehat{\mathbf{M}}$ as $\widehat{\mathbf{M}} \times \mathbf{G}$ returning explicit matrix $\in \mathbb{R}^{r \times \cdot}$.

---

[7]$\widehat{\mathbf{Q}} \leftarrow \texttt{orthonorm}(\mathbf{Q})$ yields $\widehat{\mathbf{Q}}$ such that $\widehat{\mathbf{Q}}^{\top} \widehat{\mathbf{Q}} = \mathbf{I}$ and $\texttt{span}(\widehat{\mathbf{Q}}) = \texttt{span}(\mathbf{Q})$.

**Algorithm 1** Computes rank-$k$ SVD of implicit matrix $\widehat{\mathbf{M}}$, defined symbolically as in §A.3. Follows prototype of Halko et al. [2009] but uses an alternative routine for orthonormalization.

---

1: **input:** SVD rank $k \in \mathbb{N}_+$; implicit matrix $\widehat{\mathbf{M}} \in \mathbb{R}^{r \times c}$ implementing functions in §A.3.
2: **procedure** iSVD($\widehat{\mathbf{M}}, k$)
3:     $(r, c) \leftarrow \widehat{M}.\texttt{shape}()$
4:     $\mathbf{Q} \leftarrow \sim \mathcal{N}(0, 1)^{c \times 2k}$     ▷ Random matrix sampled from standard normal. Shape: $(c \times 2k)$
5:     **for** $i \leftarrow 1$ **to** iterations **do**
6:       $\mathbf{Q} \leftarrow \texttt{orthonorm}(\widehat{\mathbf{M}}.\texttt{dot}(\mathbf{Q}))$         ▷ $(r \times 2k)$
7:       $\mathbf{Q} \leftarrow \texttt{orthonorm}(\widehat{\mathbf{M}}.\texttt{T}().\texttt{dot}(\mathbf{Q}))$     ▷ $(c \times 2k)$
8:     $\mathbf{Q} \leftarrow \texttt{orthonorm}(\widehat{\mathbf{M}}.\texttt{dot}(\mathbf{Q}))$     ▷ $(r \times 2k)$
9:     $\mathbf{B} \leftarrow \widehat{\mathbf{M}}.\texttt{T}().\texttt{dot}(\mathbf{Q})^\top$     ▷ $(2k \times c)$
10:    $\mathbf{U}, \mathbf{s}, \mathbf{V}^\top \leftarrow \texttt{tf.linalg.svd}(B)$
11:    $\mathbf{U} \leftarrow \mathbf{Q} \times \mathbf{U}$     ▷ $(r \times 2k)$
12:    **return** $\mathbf{U}[:, :k], \mathbf{s}[:k], \mathbf{V}[:, :k]$

---

**Algorithm 2** Comparison of orthonorm routines. **Left**: Halko et al. [2009] (via QR decomposition). **Right**: ours (via Cholesky decomposition).

---

| | |
|---|---|
| 1: **input:** matrix $\mathbf{Q}$. | 1: **input:** matrix $\mathbf{Q}$. |
| 2: **procedure** orthonormHalko($\mathbf{Q}$) | 2: **procedure** orthonormOurs($\mathbf{Q}$) |
| 3:   $\widehat{\mathbf{Q}}, \mathbf{R} \leftarrow \texttt{QRdecomposition}(\mathbf{Q})$ | 3:   $\mathbf{L} \leftarrow \texttt{Choleskydecomposition}(\mathbf{Q}^\top \mathbf{Q})$ |
| 4:   **return** $\widehat{\mathbf{Q}}$ | 4:   **return** $\widehat{\mathbf{Q}} \leftarrow \mathbf{Q} \times (\mathbf{L}^{-1})^\top$ |

---

3. `T()` must return an instance of `SymbolicPF` which is the (implicit) transpose of $\widehat{\mathbf{M}}$ i.e. with `.shape()` $= (c, r)$

The names of the functions were purposefully chosen to match common naming conventions, such as of numpy. We now list our current implementations of `PF` classes used for symbolically representing the design matrices discussed in the paper ($\widehat{\mathbf{M}}^{(\text{NE})}$ and $\widehat{\mathbf{M}}^{(\text{NC})}$).

### A.3.1 Leaf nodes

The leaf nodes **explicitly** hold the underlying matrix. Next, we show an implementation for a dense matrix (left), and another for a sparse matrix (right):

```
class DenseMatrixPF(SymbolicPF):
  """(implicit) matrix is explicitly
     stored dense tensor."""

  def __init__(self, m):
    self.m = tf.convert_to_tensor(m)

  def dot(self, g):
    return tf.matmul(self.m, g)

  @property
  def shape(self):
    return self.m.shape

  @property
  def T(self):
    return DenseMatrixPF(
      tf.transpose(self.m))
```

```
class SparseMatrixPF(SymbolicPF):
  """(implicit) matrix is explicitly
     stored dense tensor."""

  def __init__(self, m):
    self.m = scipy_csr_to_tf_sparse(m)

  def dot(self, g):
    return tf.sparse.sparse_dense_matmul(
      self.m, g)

  @property
  def shape(self):
    return self.m.shape

  @property
  def T(self):
    return SparseMatrixPF(
      tf.sparse.transpose(self.m))
```

### A.3.2 Symbolic nodes

Symbolic nodes **implicitly** hold a matrix. Specifically, their constructors (`__init__`) accept one-or-more other (leaf or) symbolic nodes and their implementations of `.shape()` and `.dot()` invokes those of the nodes passed to their constructor. Let us take a few examples:

```python
class SumPF(SymbolicPF):
  def __init__(self, pfs):
    self.pfs = pfs
    for pf in pfs:
      assert pf.shape == pfs[0].shape

  def dot(self, m):
    sum_ = self.pfs[0].dot(m)
    for pf in self.pfs[1:]:
      sum_ += pf.dot(m)
    return sum_

  @property
  def T(self):
    return SumPF([f.T for f in self.pfs])

  @property
  def shape(self):
    return self.pfs[0].shape
```

```python
class ProductPF(SymbolicPF):
  def __init__(self, pfs):
    self.pfs = pfs
    for i in range(len(pfs) - 1):
      assert (pfs[i].shape[1]
              == pfs[i+1].shape[0])

  @property
  def shape(self):
    return (self.pfs[0].shape[0],
            self.pfs[-1].shape[1])

  @property
  def T(self):
    return ProductPF(reversed(
      [f.T for f in self.pfs]))

  def dot(self, m, cache=None):
    product = m
    for pf in reversed(self.pfs):
      product = pf.dot(product)
    return product
```

Our attached code (will be uploaded to github, after code review process) has concrete implementations for other PFs. For example, `GatherRowsPF` and `GatherColsPF`, respectively, for (implicitly) selecting row and column slices; as well as, concatenating matrices, and multiplying by scalar. Further, the actual implementation has further details than displayed, for instance, for implementing lazy caching §A.3.4.

### A.3.3 Syntactic sugar

Additionally, we provide a couple of top-level functions e.g. `fsvd.sum()`, `fsvd.gather()`, with interfaces similar to numpy and tensorflow (but expects implicit PF as input and outputs the same). Further, we override python operators (e.g., `+`, `-`, `*`, `**`, `@`) on the base `SymbolicPF` that instantiates symbolic node implementations, as:

```python
class SymbolicPF:

  #   ....

  def __add__(self, other):
    return SumPF([self, other])

  def __matmul__(self, other):
    return ProductPF([self, other])

  def __sub__(self, other):
    return SumPF([self, TimesScalarPF(-1, other)])

  def __mul__(self, scalar):
    return TimesScalarPF(scalar, self)

  def __rmul__(self, scalar):
    return TimesScalarPF(scalar, self)

  def __pow__(self, integer):
    return ProductPF([self] * int(integer))
```

This allows composing implicit matrices using syntax commonly-used for calculating explicit matrices. We hope this notation could ease the adoption of our framework.

### A.3.4 Optimization by lazy caching

One can come up with multiple equivalent mathematical expressions. For instance, for matrices $\mathbf{A}, \mathbf{B}, \mathbf{C}$, the expressions $\mathbf{A} \times \mathbf{B} + \mathbf{A}^2\mathbf{C}$ and $\mathbf{A} \times (\mathbf{B} + \mathbf{A}\mathbf{C})$ are equivalent. However, the first one should be cheaper to compute.

At this point, we have not implemented such equivalency-finding: we do not Substitute expressions with their more-efficient equivalents. Rather, we implement (simple) optimization by *lazy-caching*. Specifically, when computing product $\widehat{\mathbf{M}}^\top\mathbf{G}$, as the `dot()` methods are recursively called, we populate a cache (python `dict`) of intermediate products with key-value pairs:

- key: ordered list of (references of) `SymbolicPF` instances that were multiplied by $\mathbf{G}$;
- value: the product of the list times $\mathbf{G}$.

While this optimization scheme is suboptimal, and trades memory for computation, we observe up to 4X speedups on our tasks (refer to Fig. 2). Nonetheless, in the future, we hope to utilize an open-source *expression optimizer*, such as TensorFlow's [Lattner et al., 2021].

### A.4 Approximating the integral of Gaussian 1d kernel

The integral from Equation 9 can be approximated using softmax, as:

$$\int_\Omega \mathbf{S}^x \overline{\mathcal{N}}(x \mid \mu, s)\, dx = \frac{\int_\Omega \mathbf{S}^x \exp\left(-\frac{1}{2}\left(\frac{x-\mu}{s}\right)^2\right)\, dx}{\int_\Omega \exp\left(-\frac{1}{2}\left(\frac{y-\mu}{s}\right)^2\right)\, dy} \approx \frac{\sum_{x\in\overline{\Omega}} \mathbf{S}^x \exp\left(-\frac{1}{2}\left(\frac{x-\mu}{s}\right)^2\right)\, dx}{\sum_{y\in\overline{\Omega}} \exp\left(-\frac{1}{2}\left(\frac{y-\mu}{s}\right)^2\right)\, dy}$$

$$= \{\mathbf{S}^x\}_{x\in\overline{\Omega}} \otimes \operatorname*{softmax}_{x\in\overline{\Omega}}\left(-(x-\mu)^2 \exp(\overline{s})\right)$$

where the first equality expands the definition of truncated normal, i.e. divide by partition function, to make the mass within $\Omega$ sum to 1. In our experiments, we use $\Omega = [0.5, 2]$. The $\approx$ comes as we use discretized $\overline{\Omega} = \{0.5, 0.505, 0.51, 0.515, \ldots, 2.0\}$ (i.e., with 301 entries). Finally, the last expression contains a constant tensor (we create it only once) containing $\mathbf{S}$ raised to every power in $\overline{\Omega}$, stored along (tensor) axis which gets multiplied (via tensor product i.e. *broadcasting*) against softmax vector (also of 301 entries, corresponding to $\overline{\Omega}$). We parameterize with two scalars $\mu, \overline{s} \in \mathbb{R}$ i.e. implying $\overline{s} = \log \frac{1}{2s^2}$

### A.5 Analysis, theorems and proofs

#### A.5.1 Norm regularization of wide models

Proof of Theorem 1 is common in university-level linear algebra courses, but here for completeness. It imples that if $\widehat{\mathbf{M}}^{(\mathrm{NC})}$ is too wide, then we need **not** to worry much about *overfitting*.

*Proof.* of Theorem 1

Assume $\mathbf{Y} = \mathbf{y}$ is a column vector (the proof can be generalized to matrix $\mathbf{Y}$ by repeated column-wise application[8]). $\mathrm{SVD}_k(\widehat{\mathbf{M}})$ for $k \geq \mathrm{rank}(\widehat{\mathbf{M}})$, recovers the solution:

$$\widehat{\mathbf{W}}^* = \left(\widehat{\mathbf{M}}\right)^\dagger \mathbf{y} = \widehat{\mathbf{M}}^\top \left(\widehat{\mathbf{M}}\widehat{\mathbf{M}}^\top\right)^{-1} \mathbf{y}. \tag{16}$$

The *Gram matrix* $\widehat{\mathbf{M}}\widehat{\mathbf{M}}^\top$ is nonsingular as the rows of $\widehat{\mathbf{M}}$ are linearly independent. To prove the claim let us first verify that $\widehat{\mathbf{W}}^* \in \widehat{\mathcal{W}}^*$:

$$\widehat{\mathbf{M}}\widehat{\mathbf{W}}^* = \widehat{\mathbf{M}}\widehat{\mathbf{M}}^\top \left(\widehat{\mathbf{M}}\widehat{\mathbf{M}}^\top\right)^{-1} \mathbf{y} = \mathbf{y}.$$

---

[8]Minimizer of Frobenius norm is composed, column-wise, of minimizers $\operatorname{argmin}_{\widehat{\mathbf{M}}\mathbf{W}_{:,j}=\mathbf{Y}_{:,j}} ||\mathbf{W}_{:,j}||_2^2, \forall j$.

Let $\widehat{\mathbf{W}}_p \in \widehat{\mathcal{W}}^*$. We must show that $||\widehat{\mathbf{W}}^*||_2 \leq ||\widehat{\mathbf{W}}_p||_2$. Since $\widehat{\mathbf{M}}\widehat{\mathbf{W}}_p = \mathbf{y}$ and $\widehat{\mathbf{M}}\widehat{\mathbf{W}}^* = \mathbf{y}$, their subtraction gives:

$$\widehat{\mathbf{M}}(\widehat{\mathbf{W}}_p - \widehat{\mathbf{W}}^*) = 0. \tag{17}$$

It follows that $(\widehat{\mathbf{W}}_p - \widehat{\mathbf{W}}^*) \perp \widehat{\mathbf{W}}^*$:

$$(\widehat{\mathbf{W}}_p - \widehat{\mathbf{W}}^*)^\top \widehat{\mathbf{W}}^* = (\widehat{\mathbf{W}}_p - \widehat{\mathbf{W}}^*)^\top \widehat{\mathbf{M}}^\top \left(\widehat{\mathbf{M}}\widehat{\mathbf{M}}^\top\right)^{-1} \mathbf{y}$$

$$= \underbrace{(\widehat{\mathbf{M}}(\widehat{\mathbf{W}}_p - \widehat{\mathbf{W}}^*))^\top}_{=0 \text{ due to Eq. 17}} \left(\widehat{\mathbf{M}}\widehat{\mathbf{M}}^\top\right)^{-1} \mathbf{y} = 0$$

Finally, using Pythagoras Theorem (due to $\perp$):

$$||\widehat{\mathbf{W}}_p||_2^2 = ||\widehat{\mathbf{W}}^* + \widehat{\mathbf{W}}_p - \widehat{\mathbf{W}}^*||_2^2$$

$$= ||\widehat{\mathbf{W}}^*||_2^2 + ||\widehat{\mathbf{W}}_p - \widehat{\mathbf{W}}^*||_2^2 \geq ||\widehat{\mathbf{W}}^*||_2^2$$

$\square$

### A.5.2   At initialization, deep model is identical to linear (convexified) model

*Proof.* of Theorem 2:

The layer-to-layer "positive" and "negative" weight matrices are initialized as: $\mathbf{W}_{(\text{p})}^{(l)} = -\mathbf{W}_{(\text{n})}^{(l)} = \mathbf{I}$. Therefore, at initialization:

$$\mathbf{H}^{(l+1)} = \left[\widehat{\mathbf{A}}\mathbf{H}^{(l)}\mathbf{W}_{(\text{p})}^{(l)}\right]_+ - \left[\widehat{\mathbf{A}}\mathbf{H}^{(l)}\mathbf{W}_{(\text{n})}^{(l)}\right]_+ = \left[\widehat{\mathbf{A}}\mathbf{H}^{(l)}\right]_+ - \left[-\widehat{\mathbf{A}}\mathbf{H}^{(l)}\right]_+$$

$$= \mathbb{1}_{\left[\widehat{\mathbf{A}}\mathbf{H}^{(l)} \geq 0\right]} \circ \left(\widehat{\mathbf{A}}\mathbf{H}^{(l)}\right) - \mathbb{1}_{\left[-\widehat{\mathbf{A}}\mathbf{H}^{(l)} \geq 0\right]} \circ \left(-\widehat{\mathbf{A}}\mathbf{H}^{(l)}\right)$$

$$= \mathbb{1}_{\left[\widehat{\mathbf{A}}\mathbf{H}^{(l)} \geq 0\right]} \circ \left(\widehat{\mathbf{A}}\mathbf{H}^{(l)}\right) + \mathbb{1}_{\left[-\widehat{\mathbf{A}}\mathbf{H}^{(l)} \geq 0\right]} \circ \left(\widehat{\mathbf{A}}\mathbf{H}^{(l)}\right)$$

$$= \left(\mathbb{1}_{\left[\widehat{\mathbf{A}}\mathbf{H}^{(l)} \geq 0\right]} + \mathbb{1}_{\left[-\widehat{\mathbf{A}}\mathbf{H}^{(l)} \geq 0\right]}\right) \circ \left(\widehat{\mathbf{A}}\mathbf{H}^{(l)}\right)$$

$$= \widehat{\mathbf{A}}\mathbf{H}^{(l)},$$

where the first line comes from the initialization; the second line is an alternative definition of relu: the indicator function $\mathbb{1}$ is evaluated element-wise and evaluates to 1 in positions its argument is true and to 0 otherwise; the third line absorbs the two negatives into a positive; the fourth by factorizing; and the last by noticing that **exactly one** of the two indicator functions evaluates to 1 almost everywhere, except at the boundary condition i.e. at locations where $\widehat{\mathbf{A}}\mathbf{H}^{(l)} = 0$ but there the right-term makes the Hadamard product 0 regardless. It follows that, since $\mathbf{H}^{(0)} = \mathbf{X}$, then $\mathbf{H}^{(1)} = \widehat{\mathbf{A}}\mathbf{X}$, $\quad \mathbf{H}^{(2)} = \widehat{\mathbf{A}}^2\mathbf{X}$, $\ldots, \mathbf{H}^{(L)} = \widehat{\mathbf{A}}^L\mathbf{X}$.

The layer-to-output positive and negative matrices are initialized as: $\mathbf{W}_{(\text{op})}^{(l)} = -\mathbf{W}_{(\text{on})}^{(l)} = \widehat{\mathbf{W}}^*_{[dl\,:\,d(l+1)]}$. Therefore, at initialization, the final output of the model is:

$$\mathbf{H} = \sum_{l=0}^{l=L} \left[\mathbf{H}^{(l)}\mathbf{W}_{(\text{op})}^{(l)}\right]_+ - \left[\mathbf{H}^{(l)}\mathbf{W}_{(\text{on})}^{(l)}\right]_+ = \sum_{l=0}^{l=L} \left[\mathbf{H}^{(l)}\widehat{\mathbf{W}}^*_{[dl\,:\,d(l+1)]}\right]_+ - \left[-\mathbf{H}^{(l)}\widehat{\mathbf{W}}^*_{[dl\,:\,d(l+1)]}\right]_+$$

$$= \sum_{l=0}^{l=L} \mathbf{H}^{(l)}\widehat{\mathbf{W}}^*_{[dl\,:\,d(l+1)]},$$

where the first line comes from the definition and the initialization. The second line can be arrived by following exactly the same steps as above: expanding the re-writing the ReLu using indicator notation, absorbing the negative, factorizing, then lastly unioning the two indicators that are mutually exclusive

almost everywhere. Finally, the last summation can be expressed as a block-wise multiplication between two (partitioned) matrices:

$$\mathbf{H} = \begin{bmatrix} \mathbf{H}^{(0)} & \vdots & \mathbf{H}^{(1)} & \vdots & \dots & \vdots & \mathbf{H}^{(L)} \end{bmatrix} \begin{bmatrix} \widehat{\mathbf{W}}^*_{[0\,:\,d]} \\ \widehat{\mathbf{W}}^*_{[d\,:\,2d]} \\ \dots \\ \widehat{\mathbf{W}}^*_{[dL\,:\,d(L+1)]} \end{bmatrix} = \begin{bmatrix} \mathbf{X} & \vdots & \widehat{\mathbf{A}}\mathbf{X} & \vdots & \dots & \vdots & \widehat{\mathbf{A}}^L\mathbf{X} \end{bmatrix} \widehat{\mathbf{W}}^*$$

$$= \widehat{\mathbf{M}}^{(\text{NC})}\widehat{\mathbf{W}}^* = \mathbf{H}^{(\text{NC})}_{\text{linearized}}$$

$\square$

### A.5.3  Computational Complexity and Approximation Error

**Theorem 3.** (Linear Time) *Implicit SVD (Alg. 1) trains our convexified GRL models in time linear in the graph size.*

*Proof.* of Theorem 3 for our two model families:

1. For rank-$k$ SVD of $\widehat{\mathbf{M}}^{(\text{NE})}$: Let cost of running $\widehat{\mathbf{M}}^{(\text{NC})}$.dot() be $T_{\text{mult}}$. The run-time to compute SVD, as derived in Section 1.4.2 of [Halko et al., 2009], is:

$$\mathcal{O}(kT_{\text{mult}} + (r + c)k^2). \tag{18}$$

Since $\widehat{\mathbf{M}}^{(\text{NE})}$ can be defined as $C$ (context window size) multiplications with sparse $n \times n$ matrix $\mathcal{T}$ with $m$ non-zero entries, then running $\text{iSVD}_k(\widehat{\mathbf{M}}^{(\text{NE})})$ costs:

$$\mathcal{O}(kmC + nk^2) \tag{19}$$

2. For rank-$k$ SVD over $\widehat{\mathbf{M}}^{(\text{NC})}$: Suppose feature matrix contains $d$-dimensional rows. One can calculate $\widehat{\mathbf{M}}^{(\text{NC})} \in \mathbb{R}^{n \times Ld}$ with $L$ sparse multiplies in $\mathcal{O}(Lmd)$. Calculating and running SVD [see Section 1.4.1 of Halko et al., 2009] on $\widehat{\mathbf{M}}^{(\text{JKN})}$ costs total of:

$$\mathcal{O}(ndL\log(k) + (n + dL)k^2 + Lmd). \tag{20}$$

Therefore, training time is linear in $n$ and $m$. $\square$

Contrast with methods of WYS [Abu-El-Haija et al., 2018] and NetMF [Qiu et al., 2018], which require assembling a dense $n \times n$ matrix requiring $\mathcal{O}(n^2)$ time to decompose. One wonders: how far are we from the optimal SVD with a linear-time algorithm? The following bounds the error.

**Theorem 4.** (Exponentially-decaying Approx. Error) *Rank-$k$ randomized SVD algorithm of Halko et al. [2009] gives an approximation error that can be brought down, exponentially-fast, to no more than twice of the approximation error of the optimal (true) SVD.*

*Proof.* of Theorem 4 is in Theorem 1.2 of Halko et al. [2009] $\square$

Consequently, compared to $\widetilde{\text{NetMF}}$ of [Qiu et al., 2018], which incurs unnecessary estimation error as they sample the matrix entry-wise, our estimation error can be brought-down exponentially by increasing the iterations parameter of Alg. 1. In particular, as long as we can compute products against the matrix, we can decompose it, almost as good as if we had the individual matrix entries.