# OpenReview forum: "Implicit SVD for Graph Representation Learning"
_NeurIPS.cc/2021/Conference — NeurIPS 2021 Poster_

### Official Review · Reviewer_L58o · 2021-07-15

**Rating:** 6
**Confidence:** 3

**Summary:**

In this work, authors design a software framework which allows symbolically representing matrices and computing their SVD implicitly. This paper also derives linear first-order approximations of popular Graph Representation Learning (GRL) models, and they can be trained via SVD. Further, SVD initializations can be used to fine-tune deeper neural networks via cross-entropy on many GRL tasks. Experimental results demonstrate that the proposed model trains orders-of-magnitudes faster than state-of-art methods, while having comparable performance.

**Limitations And Societal Impact:**

Authors have discussed the limitations and societal impacts in the Conclusion Section.

**Main Review:**

\textbf{Strength}:
Authors open-source a software library that allows symbolic definition of matrices and computing their SVD efficiently without explicitly calculating them. \\
Authors derive the convexification of popular GRL models (i.e., network embedding models and message passing graph networks), and presents how to leverage SVD of designed matrices to train them without computing gradients.\\
Convexified model parameters from SVD can initialize deeper models which can be fine-tuned via cross-entropy over GRL tasks.


\textbf{Weakness}:
First, it would be great if the authors could discuss about the proposed method's computational complexity when dealing with huge graphs.\\
Next, it seems that the experiments are conducted on simple graphs. I wonder if it is possible to extend the proposed model to more complex graphs (e.g., knowledge graphs)?


**Time Spent Reviewing:**

3.5

---

> ### Author Response · Authors · 2021-08-05
> **Discussing your pointed-out weaknesses: Scale to large graphs & handling tasks that differ from the 2 kinds we discuss in paper.**
>
> We appreciate your time for reviewing our work! The two concerns you brought are spot-on, and are also brought-up by other reviewers (great minds think alike :-)
>
> 1. **Computational Complexity & Scale to large graphs**: Please see ​​Common Concern #1 (addressing all reviewers).
>
> 2. **More Complex Graphs**. Please see Common Concern #2 (addressing all reviewers).
> Knowledge Graphs (KGs) are multigraphs containing multi-class link prediction tasks, where node have features, and edges could be temporal. Although this paper did not work-out neither a (i) convex model nor (ii) its deeper counterpart for KG, we believe such derivation is possible, though could utilize domain knowledge. KG is outside the scope of our current paper (but is an attractive next step, due to wide application of KGs). Nonetheless, we will make it clear, in this main paper, that each new model family requires some hand-derivation for both (i) and (ii).

---

> ### Comment · Reviewer_L58o · 2021-08-27
> **keep my original score**
>
> After reading authors' response, I will keep my original score.

---

> > ### Author Response · Authors · 2021-08-28
> > **Thank you :)**
> >
> > We appreciate your contributions to NeurIPS -- we are reviewers ourselves and understand the time commitment.
> >
> > After the final decision, regardless of outcome, we plan to post the paper on ArXiv, and will be expanding the text on scalability (currently in "limitations" subsection, in page 9 of draft you reviewed). We also plan to upload our code on github.
> >
> > Luckily, just yesterday, we got the distributed implementation working (and applied it to the largest OGB graphs)  -- ran on 8 machines via PyTorch RPC. We hope the scalable contribution will follow onto github, shortly after clean-up and documentation. We also plan to work on KGs, but it is just too much to fit into the same paper.
> >
> > If the above is lots of text for an appreciation, you may just read the title ("Thank you") and skip the remainder.

---

### Official Review · Reviewer_bCFd · 2021-07-16

**Rating:** 6
**Confidence:** 4

**Summary:**

The goal of this paper is to speed-up graph representational learning by removing
nonlinearities between layers and replacing them with linear models while also
replacing cross entropy by Frobenius norm minimization. In turn, this allows the
application of SVD, and the authors suggest modifications of randomized SVD
which execute faster and in a matrix-free manner. The above approach is reported
to be much faster than previous state-of-the-art, even though it is required to post-process
the obtained model by feeding it as an initial guess to a more expensive, but also
more accurate, approach.

**Main Review:**


-) The paper is fairly well-written.

-) For most part, the concepts presented are in a clear manner although the paper is
crowded with equations and side-information.

-) I am not sure about whether the authors claim that the their orthonormalization approach
is new or they just use it to speed-up the QR part in randomized SVD. If the former is the
case, then it is not a new idea. In fact, it is a well-known "trick", see for example

https://www.sciencedirect.com/science/article/pii/S0010465510000524

Do the authors use a different approach? If so, can they elaborate more on how exactly they
replace standard QR?

-) In addition to the above remark, using the Cholesky factorization squares the condition
number of the matrix whose columns are orthonormalized (i.e., normal equations). In fact,
I am actually surprised that the authors didn't mention this, since my understanding is that
they use the "subspace iteration" version of randomized SVD.

-) Moreover, matrix-free SVD is not a new thing -- in fact several approaches exist, see for example

https://academic.oup.com/bioinformatics/article/35/19/3679/5430929?login=true

I understand that the authors refer to their own approach of doing this, but some citations
on the topic need be provided.

-) The experiments paint a nice picture, and they indeed show that the suggested hybrid
approach is a good alternative. Nonetheless, my impression is that the proposed hybrid
approach is more costly in terms of memory. Is this the case? If so, then I wonder
whether the proposed approach is useful for large graphs with millions/tens of millions of
nodes. Note that using less nodes with more edges is not as important as very large and very
sparse graphs. What is the execution time then? Can the authors provide experiments on
graphs with millions of nodes? Alongside memory costs for each method? This would be
very helpful.

-) How do you set the number of leaf nodes?

-) A '.' is missing at the end of line 285.

-) "decompose exactly M without calculating it" -> "decompose M exactly without explicitly calculating it."


**Time Spent Reviewing:**

4

---

> ### Author Response · Authors · 2021-08-05
> **Crowdedness, orthonormalization, matrix-free SVD, scale**
>
> * **Clarity & Crowdedness**: Thank you for the compliments on the writing clarity. As for the crowdedness, it is maybe somewhat hard to avoid -- we describe (i) a software framework and (ii) applications on two classes of GRL models. (i) is to build symbolic expressions of matrices (represented as DAGs) and mention that their SVD can be computed due to [Halko et al 2009] -- as multiplications against DAG nodes can be recursively propagated to leaf nodes and (ii) specializes the framework. For each model class, we derive a convexified model and its deeper counterpart.
>
> * **Orthonormalization**: The orthonormalization approach is not new (we did not claim it as one of our 3 contributions). Our implementation can be classified as a **Randomized** Subspace Iteration method, as it follows [Halko et al 2009]. We completely replace the QR by Cholesky decomposition (You may refer to Appendix, Algorithm 2). Before the replacement, running profiling tools showed that the vast majority of the processing time was spent during the QR step. Reading into [Halko et al 2009]: we understood that **any orthonormalization** routine should work fine (need not to be QR, as recommended in Halko). We do not care if Cholesky changes the condition number of the orthonormalized matrix, so long its columns are orthonormal and span the subspace of the original matrix columns (that was orthonormalized). We are happy to mention this change to the condition number (we were not aware of it, would you have a reference?). The reference you mention [Bekas & Curioni, 2010] uses Cholesky **within** Gram-Schmidt orthonormalization, which indeed could be useful for us if we do SVD with large rank -- in our applications, our largest rank was 256. However, we will add your recommended citation [Bekas & Curioni, 2010], as means of extending our methods to super-large graphs (e.g. billions of nodes & edges) and/or when larger SVD ranks were desired, which also fits well with the scale concern, pointed out the all reviewers, addressed on top.
>
> * **Matrix-Free SVD**: You are right -- we are not the first to do SVD (or PCA) on a matrix without explicitly computing it entry-wise. **We will add citation to [Bose et al 2019]**, as indeed according to their supplementary material, they do not compute the covariance matrix $A A^\top$, but rather, they compute its product against $Q$ as $A ( A^\top Q)$. In fact, this is a property of the underlying algorithm prototype outlined in [Halko, 2009] (need not to compute the actual matrix, entry-wise, as we mention on **line 151** in our paper). Our contribution here is to allow flexibility in constructing (designing) the matrices by overloading the operators (e.g., @, +, -) and defining functions (e.g., concatenation, dropout) -- of course, such composition and overloading of operators exists in math frameworks, such as TensorFlow or PyTorch, but it has not been applied to build implicit matrices to run SVD (or PCA), as far as we are aware.
>
> * **Scale to large sparse graphs**: Please see ​​Common Concern #1 (addressing all reviewers).
>
> * **Typos** we just fixed the two typos, in our local paper version, as you recommended -- thank you!
>
> * **Answer to: How do you set the number of leaf nodes?** We apologize about the confusion of naming. Node, here, refers to a *computation node that holds a matrix* i.e. on the computational DAG, not a node on the graph. Specifically, as shown in Figure 1, each leaf node corresponds to a data matrix (e.g., Adjacency matrix, Transition matrix, Feature matrix, etc). The number of leaf nodes = the number of data matrices (e.g., adjacency), that the practitioner decides to load in order to symbolically define the design matrix M, such that svd(M) yields parameter values to the convexified model.

---

### Official Review · Reviewer_NLBE · 2021-07-19

**Rating:** 9
**Confidence:** 4

**Summary:**

The authors suggest to express large matrices as operations over given manageable leaf matrices. They keep track of the resulting abstract syntax tree (AST) and offer a specialised SVD decomposition implementation that can perform the computation on the final expression by traversing the AST and operating on the individual leaf matrices. They then show how using the SVD operation they can perform learning in various inductive and transductive tasks. The advantage of their proposal is that the process is computationally efficient and they can in practice match SOTA results with speedups reaching several orders of magnitude.

**Limitations And Societal Impact:**

The authors do address the limitations and potential negative societal impact of their work.

**Main Review:**

The work presented warrants dissemination as it is an original approach to tackle ML problems on large datasets.
The paper is filled with ideas and examples on how to model various problems and it may therefore cram too much in too little space. This is likely inevitable and not something that can be helped much.
Further improvements would include a more systematic empirical investigation of the scaling behaviour of the method (i.e. tackling a defined set of problems on ever increasing inputs)  and offering a more structured conceptual framework to describe general classes of problems and how they can be addressed/converted with the new approach.

### After the authors' responses
Many thanks to the authors for the their response. I maintain my positive evaluation. I understand that due to the space constraints the authors cannot further develop general guiding principles or offer a more systematic treatment of their proposed approach, but I am confident that they will use other media (blogs, websites, GitHub, etc) and further publications to expand on the ideas they have introduced.

**Time Spent Reviewing:**

2

---

> ### Author Response · Authors · 2021-08-05
> **Both of your comments are addressed in Common Concerns**
>
> Thank you for your feedback and for raising crucial discussion points, especially for raising the need for a *“structured conceptual framework”* that can convert existing model classes to the current approach. We addressed this point, as well as the scalability point, in Common Concerns. We will acknowledge your (anonymous) name, in the main paper. We are happy to phrase this as an open-question in the write-up (how to come up with a conceptual framework), if you see fit.

---

### Author Response · Authors · 2021-08-05
**Common concerns & Thank you!**

# Thank you
We feel the common consensus is that the reviewers are excited about our work. We are excited about it, too! Thank you! More importantly, we appreciate the time you put in, thus far, for reviewing for NeurIPS’21 -- your efforts are invaluable for the academic community.

Many of your comments have already been incorporated into our (local) draft of the paper, and the remainder will be incorporated over the next few days.

# Common Concerns
Reviewers have agreed upon these two concerns

1. Would our method scale to larger inputs? e.g. larger adjacency and/or larger feature matrix.
2. What if the "task type" is different from what has been presented? Can our method be utilized for new task types?

We will address these two concerns here, and we will also address individual reviewer comments independently.


## Common Concern #1: Scalability of Method

Computation and memory complexity of our method is **linear** in input size O(#edges + #nodes), for both discussed classes of GRL models. See Appendix (A.5.3) Theorem 3. Therefore, as long as the "leaf objects" (e.g., adjacency or feature matrix) **fit in main memory**, our current implementation scales to the task. **However**, if the "leaf objects" are **larger than main memory** (e.g., billions of nodes and/or edges), as we admit in **Limitations of our work** in the Conclusion section, then our current implementation would fail to load the leaf objects into memory. Nonetheless, extending our implementation to these settings is possible by relying on distributed implementations of matrix multiplications, although it requires some engineering efforts. The core of our implementation propagates, down the computational DAG, matrix multiples towards leaf objects. Integrating our implementation with libraries, e.g.,  MapReduce, Spark, or Apache Mahout, that implement distributed matrix multiplications, is an attractive next step, but is outside this paper's scope. This distributed engineered extension is part of our ongoing work.

## Common Concern 2: Task type -- conceptual framework for converting general classes of model families into our framework

In our write-up, we presented 2 kinds of GRL tasks, and their SOTA model families. (1) (binary) link prediction, when nodes have no features [with skipgram-like models] and  (2) node classification, when nodes have features [with GCN-like models]. These tasks are fairly popular (heavily cited) or are part of Stanford’s (new, but quickly-adopted) OGB datasets. The question of "porting" a new task type into our framework can be broken into two steps: (i) hand-deriving a convexified model and (ii) its deeper counterpart, that is equivalent to (i) at the initialization point. In addition to the two task types in the paper, we have done this exercise for one more task type (we did not include in the paper due to space constraints, but happy to include it in the additional page, if you see fit)-- we obtain SOTA results on Stanford’s ogbl-collab, which is a (binary) link prediction, when nodes have features.

Nonetheless, the general question stands: is there a systematic way of fitting a new task into our method?

Designing (i) a convexified model (via an implicit design matrix) stems from intuition. As reviewed in Section 2.1, Skipgram-like models (in expectation) regresses$^{[1]}$ to a square matrix (with $\mathcal{O}(n^2)$ non-zeroes) $^{[1]}$: minimizing cross-entropy. Our convexification of Skipgram models stems directly from the square matrix (positive term - $\lambda \times$ negative term). GCN-like models contain GCN layer ($Y=\sigma(AXW)$) and here we removed non-linearities between layers and concatenate all layers at the output. For new tasks, there could be other intuitions that practitioners may follow.

For (ii), designing a deep network that could be initialized from (i) is also a creative process. It is not clear if there exists a systematic way for this process (though it is an interesting question). However, *some elements of our work might be useful*. For instance, ReLu activation is a popular nonlinearity in Deep Networks, our split ReLu trick (see Section 5.2) can turn any **linear layer** $y = Wx$, with weight $W$ into a ReLu layer:

$y = \sigma(W_1 x) - \sigma(W_2 x) $


where $\sigma$ is ReLu and subtraction $-$ is element-wise. Indeed, the new non-linear layer is identical to the linear layer at initialization:

$W_1 = -W_2 = W$

After initialization, gradient updates w.r.t. $W_1$ and $W_2$ are expected to diverge $W_1$ from $-W_2$, as (almost always) exactly one of $W_{1ij}$ or $W_{2ij}$ will receive a non-zero gradient, for all {i, j}, at that initialization point.

Overall, although we do not have a conceptual systematic proposal for turning any model into our framework, we feel there might be multiple such systematic proposals, where each could be geared towards different kind of models (e.g. (Graph)ResNet VS DenseNets VS attention VS different activations than ReLu .. etc)

---

### Decision · Program_Chairs · 2021-09-27

**Decision:**

Accept (Poster)

**Comment:**

The reviewers were generally positive about this paper -- they liked the SVD based approach to simplifying or initializing deep neural networks for graph representation learning, and appreciated the positive empirical results. Thus, the paper is recommended for acceptance. Connections to prior work should be clarified in the revise submission -- the abstract of the paper seems to imply that the idea of computing a partial SVD of an implicit matrix is new. There is significant work in the numerical linear algebra community on matrix-free SVD methods -- in fact this is a critically important feature of nearly all Krylov based and sketching methods for partial SVD. See this link for a number of references to background literature: https://stats.stackexchange.com/questions/159325/what-fast-algorithms-exist-for-computing-truncated-svd